# Anti-Inflammatory Polymeric Nanoparticles Based on Ketoprofen and Dexamethasone

**DOI:** 10.3390/pharmaceutics12080723

**Published:** 2020-07-31

**Authors:** Eva Espinosa-Cano, Maria Rosa Aguilar, Yadileiny Portilla, Domingo F. Barber, Julio San Román

**Affiliations:** 1Biomaterials Group, Institute of Polymer Science and Technology (ICTP-CSIC), 28006 Madrid, Spain; e.espinosa.cano@csic.es (E.E.-C.); jsroman@ictp.csic.es (J.S.R.); 2Networking Biomedical Research Centre in Bioengineering, Biomaterials and Nanomedicine (CIBER-BBN), 28029 Madrid, Spain; 3Department of Immunology and Oncology, and NanoBiomedicine Initiative, Spanish National Center for Biotechnology (CNB-CSIC), 28049 Madrid, Spain; yportilla@cnb.csic.es (Y.P.); dfbarber@cnb.csic.es (D.F.B.)

**Keywords:** nanoparticles, ketoprofen, dexamethasone, inflammation, macrophages, M1 and M2 markers, Il12-p40 subunit

## Abstract

Polymeric nanoparticles that combine dexamethasone and naproxen reduce inflammation and synergistically inhibit Interleukin-12b (*Il12b*) transcription in macrophages. This effect can be the result of a cyclooxygenase-dependent or a cyclooxygenase-independent mechanism. The aim of this work is to obtain potent anti-inflammatory polymeric nanoparticles by the combination of dexamethasone and ketoprofen, one of the most efficient cyclooxygenase-inhibitors among non-steroidal anti-inflammatory drugs, with appropriate hydrodynamic properties to facilitate accumulation and co-release of drugs in inflamed tissue. Nanoparticles are spherical with hydrodynamic diameter (117 ± 1 nm), polydispersity (0.139 ± 0.004), and surface charge (+30 ± 1 mV), which confer them with high stability and facilitate both macrophage uptake and internalization pathways to favor their retention at the inflamed areas and lysosomal degradation and drug release, respectively. In vitro biological studies concluded that the dexamethasone-loaded ketoprofen-bearing system is non-cytotoxic and efficiently reduces lipopolysaccharide-induced nitric oxide release. The RT-qPCR analysis shows that the ketoprofen nanoparticles were able to reduce to almost basal levels the expression of tested pro-inflammatory markers and increase the gene expression of anti-inflammatory cytokines under inflammatory conditions. However, the synergistic inhibition of *Il12b* observed in nanoparticles that combine dexamethasone and naproxen was not observed in nanoparticles that combine dexamethasone and ketoprofen, suggesting that the synergistic trans-repression of *Il12b* observed in the first case was not mediated by cyclooxygenase-dependent pathways.

## 1. Introduction

Interleukin-12 (IL12) and interleukin-23 (IL23) have recently emerged as therapeutic targets in the treatment of autoimmune/inflammatory diseases and chronic inflammatory diseases in which the T cell dominates as the primary dysfunctional cells [1,2,3]. IL12 and IL23 are mainly produced by antigen-presenting cells like macrophages and dendritic cells, and they play a key role in naïve T-cells differentiation to Th1 and Th17 cells, respectively [4]. Their combined inhibition has demonstrated potential in the treatment of a wide range of autoimmune/inflammatory diseases [5,6,7,8]. In 2000, it was discovered that IL12 and IL23 share the IL12-p40 subunit [9], and since then, its inhibition has become of therapeutic relevance. In fact, the FDA has recently approved Sterala (ustekinumab), a monoclonal antibody against this IL12-p40 subunit, for the treatment of Crohn’s disease, psoriasis, psoriatic arthritis, and plaque psoriasis [8,10]. In a previous work, our group demonstrated that the nanoparticles (NPs) combining naproxen (NAP) and dexamethasone (Dx) have a synergistic effect on the repression of *Il12b* transcript levels, the gene that codifies for IL12-p40 in macrophages [11]. However, the mechanism for this synergistic inhibition was not elucidated. Currently, it is widely accepted that beneficial effects of non-steroidal anti-inflammatory drugs (NSAIDs) are mainly mediated by cyclooxygenase (COX)-dependent mechanisms but also via COX-independent pathways [12]. In fact, NSAIDs are known to exert anti-inflammatory effects through COX inhibition, ERK, JNK and p38 MAPK pathways, and 5-lipoxygenase (5-LO) pathways, among others [13]. There are reports of modulation of *Il12b* expression in antigen-presenting cells (i.e., macrophages and dendritic cells, mainly) through all the aforementioned pathways [13,14,15,16,17]. NSAIDs as potent COX inhibitors, reduce the metabolism of arachidonic acid (AA) through the COX-dependent pathway reducing production of prostaglandins. However, accumulation of AA leads to increased metabolism through the 5-LO alternative pathway [18]. On the other hand, Dx inhibits phospholipase A2 (PLA2), the enzyme that mediates the production of AA from membrane phospholipids, reducing AA accumulation and both COX and 5-LO metabolic pathways [19]. Furthermore, regarding COX-independent mechanisms, Dx has been demonstrated to inhibit *Il-12p40* expression in LPS-stimulated human monocytic cells by down-regulating the activation of JNK MAPK pathway [20]. According to that, NSAIDs/Dx synergistic suppression of *Il12b* expression could be the result of a COX-dependent or COX-independent mechanism. This work aims to prepare a new family of nanoparticle systems that combine physically entrapped Dx and covalently linked ketoprofen (KT), a well-known NSAID presenting several orders of magnitude lower IC_50(COX)_ than NAP [21,22], in order to obtain a system with a potent anti-inflammatory capacity and to elucidate if the COX-dependent mechanism plays a key role in the synergistic *Il12b* repression observed for the Dx/NAP system [11]. For this purpose, a methacrylic derivative of KT was prepared and copolymerized with 1-vinylimidazole (VI) by free radical polymerization obtaining a pseudo-gradient microstructure that enables self-assembly into NPs in aqueous media. The stability of the system and most suitable storage conditions were evaluated as a function of concentration, pH, final volume, and freeze-drying. Hydrodynamic properties and surface charge of the NPs were optimized in order to favor macrophage uptake and the endocytic pathways for lysosomal targeting. Macrophage rapid uptake will favor the enhanced leaky vasculature and inflammatory cells sequestration (ELVIS) effect [11], and lysosome was considered the most appropriate organelle to favor enzymatic degradation and pH-mediated co-delivery of the two drugs incorporated in the NPs [23]. Dx encapsulation efficiency was also evaluated by HPLC. Finally, biological activity of the system was studied by determining cytotoxicity, changes in the expression of LPS-induced inflammatory marker genes (i.e., *Il12b*, *Il23a,* and *Tnfa*), and anti-inflammatory marker genes (i.e., *Vegfa*, *Tgfb1*, and *Il10*) in RAW264.7 macrophages.

## 2. Materials and Methods

### 2.1. Synthesis and Characterization of Ketoprofen-Derived Monomer (HKT) 

The methacrylic derivative of ketoprofen (HKT) (Figure 1a) was synthesized via esterification reaction as previously described for its homologous based on ibuprofen and naproxen [24,25]. Briefly, KT (TCI, 1 equiv) was dissolved in dichloromethane (DCM, Aldrich, St. Louis, MO, USA). Then, purified 2-hydroxyethyl methacrylate (HEMA, Aldrich, 1 equiv) and 4-dimethylaminopyridine (DMAP, Aldrich, 0.1 equiv) were added, and dicyclohexylcarbodiimide (DCC, 1 equiv. Fluka, Buchs, Switzerland) was slowly dropped into the reaction mixture under magnetic stirring and inert atmosphere (N_2_ (g)). The reaction took place overnight, under continuous stirring, and at room temperature. The resultant mixture was filtered to eliminate *N*,*N*′-diciclohexylurea (DCU) salt, and, subsequently, washed with water and a sodium bicarbonate-saturated solution (NaHCO_3_, Panreac, Barcelona, Spain). Saturated NaCl (Panreac) and anhydrous magnesium sulphate (MgSO_4_, Qemical, Esparraguera, Spain) were used to dry the final solution before DCM evaporation under reduced pressure. The structure and purity of the product were determined by proton nuclear magnetic resonance (^1^H-NMR) in a Varian Mercury 400 MHz equipment using deuterated chloroform (CDCl_3_, Aldrich) as a solvent at 25 °C.


***HKT:***
*^1^H NMR (400 MHz, CDCl_3_) δ_H_ 7.76–7.31 (m, 9H), 5.96–5.91 (s, 1H), 5.45 (s, 1H), 4.35–4.17 (m, 4H), 3.76 (q, J = 7.2 Hz, 1H), 1.80 (s, 3H), 1.47 (d, 3H).*


### 2.2. Synthesis and Characterization of Copolymer of Ketoprofen-Based Methacrylic Monomer and 1-vinyl imidazole, poly(HKT-co-VI) 

The copolymer based on HKT and 1-vinylimidazole (VI, Aldrich) was prepared via free radical polymerization (Figure 1b) and a feed molar content in HKT (F_HKT_) of 0.4, and an initial monomers concentration ([M]) of 0.5 M were used. In summary, HKT and VI were dissolved in dimethylsulfoxide (DMSO, Scharlau, Barcelona, Spain) at a concentration of 0.5 M, and after 10 min of deoxygenation with N_2_ (g), 2,2′-azobisisobutyronitrile (AIBN, 1.5 × 10^−2^ M, Merck, Kenilworth, NJ, USA) was added. After 12 h at 60 °C, the copolymerization resultant mixture was dialyzed (Spectrum Laboratories, 3.5K molecular weight cut-off) against distilled water for 72 h, and the copolymer was isolated by freeze-drying as a white powder. Reactivity ratios of HKT and VI were studied by in situ ^1^H-NMR monitorization (see Appendix A for further details).


***poly(HKT-co-VI)(48:52) (HKT48):***
*^1^H NMR (400 MHz, DMSO-d6) δ_H_ 7.95–6.45 (m, 12H (3VI + 9HKT)), 4.53–3.39 (m, 5H (5HKT)), 3.22–2.73 (s, 1H (1VI)), 2.28–0.10 (m, 10H (2VI + 8HKT))*


#### 2.2.1. Proton Nuclear Magnetic Resonance (^1^H-NMR) 

^1^H-NMR was performed in a Varian Mercury equipment operating at 400 MHz. Spectra were recorded by dissolving samples in deuteraded DMSO (DMSO-d6) at 25 °C. Copolymer composition was calculated using MestreNova 9.0 from the ^1^H-NMR integral between 7.92–6.47 ppm corresponding to the aromatic protons of both monomers and the integral between 2.28–0 ppm, which corresponds to protons of the methyl groups k and o of HKT and protons 1 and n from the main hydrophobic carbon chain of VI and HKT, respectively (Figure 1b).

#### 2.2.2. Size Exclusion Chromatography (SEC) 

HKT-based copolymer apparent average molecular weight (M_n_ and M_w_) and polydispersity index (Ð) were determined by SEC, using a Perkin-Elmer Isocratic LC pump 250 coupled to a refraction index detector (Series 200). Two Resipore columns (250 mm × 4.6 mm, Varian, Palo Alto, CA, USA) were used as solid phase, degassed chromatographic-grade dimethylformamide (DMF, 0.7 mL/min, Scharlau, Barcelona, Spain) with LiBr (0.1% *w*/*v*) was used as eluent, and temperature was fixed at 70 °C. Monodisperse PMMA standards (Scharlau) with molecular weights between 10,300 and 1,400,000 Da were used to obtain the calibration curve. Data were analyzed using the Perkin-Elmer LC solution program.

#### 2.2.3. Differential Scanning Calorimetry (DSC) 

Glass transition temperature (T_g_) was determined by differential Scanning Calorimetry (DSC) with a Perkin Elmer DSC8500 interfaced to a Pyris thermal analysis data system. Dried samples (3–5 mg) were placed in aluminium pans and heated from −20 to 120 °C at a constant speed of 20 °C/min. T_g_ was taken as the midpoint of the heat capacity transition.

### 2.3. Preparation and Characterization of Self-Assembled Nanoparticles

#### 2.3.1. Nanoprecipitation Method 

Poly(HKT-*co*-VI)-based NPs were prepared via nanoprecipitation method. Concisely, an organic solution (acetone (Scharlau): ethanol (Scharlau), 80:20 (*v*/*v*)); of the copolymer (10 mg/mL) was added dropwise to an aqueous buffer solution at pH 4 (0.1 M acetic acid and 0.1 M NaCl), a pH below the reported pK_b_ of VI (i.e., 5.5–6.1) [26]. The remaining organic solvent was eliminated by evaporation under continuous stirring overnight, and the resultant NPs were stored at 4 °C.

#### 2.3.2. Characterization of NPs 

Hydrodynamic properties were optimized as a function of final volume (V_F_ = 10 mL, 20 mL and 30 mL) and final concentration ([NPs]_F_ = 1 mg/mL and 5 mg/mL), and the evolution of hydrodynamic properties of NPs was evaluated as a function of pH and time after freeze-drying and resuspension (Appendix A). Particle size distribution and zeta potential (ξ) were determined by dynamic light scattering (DLS) and laser Doppler electrophoresis (LDE), respectively, using a Malvern Nanosizer NanoZS Instrument equipped with a 4 mW He-Ne laser (λ = 633 nm) at a scattering angle of 173°. Measurements were performed at 25 °C. For each sample, the statistical average and standard deviation of data were calculated from 3 measurements of 11 runs each, one in case of hydrodynamic diameter (Dh) and polydispersity (PdI) and 3 measurements of 20 runs each, one in case of ξ. SEM analysis of KT NPs was performed with a Hitachi SU8000 TED, cold-emission FE-SEM microscope working with an accelerating voltage 1 kV-D (see Appendix A for more details).

#### 2.3.3. Dexamethasone and Coumarin-6 Encapsulation 

Dexamethasone (Dx, Aldrich, ≥98% pure; CAS Number: 50-02-2)-loaded NPs and coumarin-6 (c6)-loaded NPs were prepared by the described nanoprecipitation method with slight modifications. Dx (5%, 10%, 15%, or 20% *w*/*w* with respect to the polymer) or c6 (Aldrich, 1% *w*/*w* with respect to the polymer), and the corresponding polymer were dissolved in a mixture of acetone:ethanol (80:20, *v*/*v*) and slowly dropped into the aqueous buffer solution (0.1 M Acetic Acid, 0.1 M NaCl) at pH 4 under magnetic stirring. NPs (3 mg/mL) were dialyzed against the same buffer for 72 h to eliminate remaining organic solvents and the soluble non-entrapped Dx or c6. The resultant NPs were filtered through 1 μm Nylon filters (Whatman Puradisc) to eliminate insoluble Dx or c6

#### 2.3.4. Encapsulation Efficiency (%EE) and Loading Capacity (%LC) 

The powder resulting from freeze-drying of Dx-loaded NPs and c6-loaded NPs was dissolved in 2 mL of acetone:ethanol (80:20, *v*/*v*). This led to the disassembly of nanoparticle structure and release of the encapsulated drug. After organic solvent evaporation overnight, the copolymer-Dx mixture and the copolymer-c6 mixture were dissolved in acetonitrile:water (80:20, *v*/*v*) or ethanol, respectively, to precipitate the polymer. Centrifugation at 10,000 rpm for 5 min at RT separated the NPs pellet from the Dx and c6-containing supernatant, which were analyzed by HPLC (Dx, λ_abs_ = 239 nm) and UV spectrophotometry (c6, λabs = 459 nm), correspondingly. Encapsulation efficiency (%EE) was computed using Equation (1) and the loading capacity (%LC) using Equation (2). According to this, NPs were labeled as XY-KT NPs being X the encapsulation efficacy and Y the drug or dye encapsulated.
(1)$$Encapsulation\;Efficiency\;\left(\%\right)=\frac{{\left[Dx/c6\right]}_{measured}}{{\left[Dx/c6\right]}_{initial}}\times 100$$
(2)$$Loading\;capacity\;\left(\%\right)=\frac{mass\;of\;Dx/c6\;measured}{mass\;of\;NPs}\times 100$$

### 2.4. Cell Culture

RAW264.7 murine macrophages (Sigma-Aldrich) were cultured in high-glucose Dulbecco’s modified Eagle’s medium (DMEM; Sigma, Saint Louis, MO, USA) supplemented with 10% (*v*/*v*) fetal bovine serum (FBS; Gibco, BRL), 2% (*v*/*v*) L-Glutamine (Sigma, Saint Louis, MO, USA), and 1% (*v*/*v*) Penicillin-G (Sigma, Saint Louis, MO, USA) at 37 °C, 5% CO_2_, and 90% relative humidity.

#### 2.4.1. Uptake Rate of c6-Loaded KT NPs by RAW264.7 Macrophages 

RAW264.7 cells were seeded into 6-well plates (1.7 × 10^5^ cells/mL) in complete DMEM. The cells were incubated overnight. The medium was replaced with the corresponding c6-loaded KT NPs suspension in DMEM (NPs:DMEM (1:5, *v*/*v*), Appendix A) that was added. Cells were incubated with NPs over different times 1, 2, 4, 6, 8, and 24 h at 37 °C. At each time point, cells were washed with cold PBS, harvested, and counted to normalize fluorescence/cell. Then, they were centrifuged at 10,000 rpm, the supernatant was discarded, and cell’s pellet was lysed with ethanol. At this point, ethanol compromised the nanoparticle structure releasing and dissolving internalized c6. After further centrifugation at 10,000 rpm, fluorescence of supernatant was measured (458/540 nm, excitation/emission) by a Multi-Detection Microplate Reader Synergy HT (BioTek Instruments; Winooski, VT, USA).

#### 2.4.2. Route of Nanoparticle Internalization 

RAW264.7 macrophages, cells were preincubated with chlorpromazine (CHL; 50 μM, Sigma-Aldrich) to inhibit clathrin-mediated endocytosis, nystatin (NYST; 10 μg/mL, Sigma-Aldrich) to inhibit caveolae-mediated endocytosis, and amiloride (AMI; 100 μM, Sigma-Aldrich) to inhibit macropinocytosis. After 30 min at 37 °C, the inhibitor solutions were removed, and freshly prepared c6-loaded KT NPs in medium (NPs:DMEM (1:5, *v*/*v*)) were added (0.25 mg/mL) and further incubated for another 8 h. Subsequently, the cells were washed and lysed as previously described. The groups treated with c6-loaded NPs but without inhibitor at 4 °C or at 37 °, were used as negative and positive control, respectively. The percentage uptake after treatment with the inhibitors was normalized to the positive control uptake, which was expressed as 100% (Equation (3)).
(3)$$\%Uptake=\frac{\left[c6\right]inhibitor}{\left[c6\right]control}x100$$

#### 2.4.3. In Vitro Cytotoxicity Assay of NPs 

In a 96-well plate under permissive conditions, 2 × 10^5^ live cells/mL (100 µL/well) were seeded. After 24 h, cells were treated with different concentrations of NPs suspension (0.250, 0.125, 0.090, 0.045, 0.023, or 0.011 mg/mL; NPs:DMEM (1:5, *v*/*v*)), and after 24 h, cell viability was determined by AlamarBlue (Invitrogen) assay. Absorbance at 570 nm was measured using a Multi-Detection Microplate Reader Synergy HT (BioTek Instruments, Winooski, VT, USA). The treatments were done in replicates (n = 8). Results were expressed as % of cell viability with respect to the control (cells treated with medium).

#### 2.4.4. Nitric Oxide (NO) Assay 

RAW264.7 macrophages were seeded in a 96-well plate (2 × 10^5^ live cells/mL, 100 µL/well). After 24 h, cells were treated with lipopolysaccharide (LPS; CAS Number: 297-473-0, Sigma-Aldrich; 5 μg/mL) and with different [NPs]_F_ of KT NPs, 14Dx-KT NPs, or free Dx. After 24 h and 48 h of treatment, NO released by macrophages was determined using Griess reagent modified kit (Sigma-Aldrich) according to the manufacturer instructions. The treatments were done in replicates (n = 8), and results were expressed as mean ± standard deviation of the percentage of NO released with respect to the control (LPS-activated cells with no further treatment or inflammatory conditions untreated (IC,U)).

#### 2.4.5. RNA Extraction, Reverse Transcription, Real-Time Quantitative PCR (RT-qPCR) 

The transcript levels of M_1_- (*Il12b*, *Il23a*, and *Tnfa*) and M_2_- (*Tgfb1*, *Il10,* and *Vegfa*) related genes were determined by quantitative RT-PCR. RAW264.7 cells were incubated 24 h with culture medium (non-inflammatory conditions, NIC) or with LPS (5 μg/mL) to simulate inflammatory conditions (IC,U), and either non-treated or treated with unloaded KT-NPs (0.045 mg/mL; NPs:DMEM (1:5, *v*/*v*)), 14Dx-KT NPs (5.1 μM Dx and 0.045 mg/mL NPs; NPs:DMEM (1:5, *v*/*v*)), or free Dx (5.1 μM). Media were collected after 24 h of treatment to eliminate non-internalized NPs or free Dx, and cells were further incubated in culture medium up to 7 days. Culture medium was refreshed every 48 h. Total RNA was extracted from cells after 1 and 7 days of NPs addition. PureLink RNA Mini Kit (Applied Biosystems, Foster City, CA, USA) was used for this purpose following manufacturer’s instructions [27]. RNA concentration was quantified by measuring absorbance at 260 and 280 nm in a NanoDrop 1000 spectrophotometer (Thermo Scientific) and 40 ng RNA/sample were transformed into cDNA using a MultiScribe reverse transcription-based reaction kit (Applied Biosystems) in the presence of an RNAse inhibitor (N8080119, Applied Biosystems) in a MyCycler thermocycler (Bio-Rad; with the following temperature profile: 25 °C—10 min, 37 °C—2 h, 85 °C—5 min, 4 °C—∞). Appendix A shows the list of specific primers used for quantitative PCR (all from Sigma). The reaction was performed using the Power SYBR Green PCR Master Mix (Applied Biosystems), in an ABI PRISM 7900HT Real-Time PCR System (Applied Biosystems with the following temperature profile: 95 °C—15 s, 60 °C—60 s, 40 cycles). Melting curves were generated in order to verify the specificity of the amplification (15 s, from 60 °C to 95 °C). RT-qPCR expression data were analyzed according to the 2^−ΔΔ*C*t^ method (Livak et al., 2001) or as 2^−Δ*C*t^ normalized to β-actin, and array views were generated using MeV software.

## 3. Results and Discussion

### 3.1. Methacrylic Derivative of Ketoprofen Monomer (HKT)

A methacrylic derivative of ketoprofen was synthesized with yields above 90% and high purity as confirmed by ^1^H-NMR spectroscopy (Figure 1a). KT was linked through the carboxylic group, the main contributor to gastrointestinal adverse effects, to HEMA, forming an ester bond which is susceptible to hydrolysis under acidic conditions and/or by esterases. pH values between 5.5 and 6.0 and high esterase concentration are encountered in the lumen of lysosomes [28]. Altogether, this may provide a pH/enzyme-accelerated release of KT at inflamed areas where pH values are around 6.4 [29], or in the lumen of lysosomes after sequestration by inflammatory cells.

### 3.2. Synthesis and Characterization of Copolymer of Ketoprofen-Based Methacrylic Monomer and 1-Vinyl Imidazole, Poly(HKT-co-VI)

The disappearance of the ^1^H-NMR vinyl proton signals of HKT and VI (CH_2-VI_ between 4.5 and 5.0 ppm and CH_2-HKT_ between 5.0 and 6.0 ppm), the new ^1^H-NMR signals resultant from the methylene protons of the backbone chains (CH_3-m_, CH_2-l_, and CH_2-n_ between 0.1 and 2.8 ppm), and the broadening of the signals as a result of the macromolecular nature of the copolymer (Figure 1b) confirmed the successful co-polymerization, at established initial conditions (F_HKT_ = 0.4 and [M] = 0.5 M), of hydrophobic HKT, and hydrophilic VI was carried out by free radical polymerization (yield = 84%).

The copolymers molar composition was quantitatively determined from their corresponding ^1^H-NMR spectra by considering the signals between 0.1 and 2.8 ppm assigned to eight protons of HKT (CH_3-k_, CH_3-o_, and CH_2-n_) and two protons of VI (CH_2-1_) and the signals between 6.5 and 8.0 ppm resultant from nine aromatic protons of HKT (CH_-a-i_) and three aromatic protons of VI (CH_-3,4,5_). The differences among copolymer HKT molar content (F_HKT_) and feed HKT molar content (f_HKT_) are explained by the two orders of magnitude difference in the reactivity ratios of the monomers (Appendix A) and the fact that total conversion was not reached. The molecular weight (M_w_) of the copolymer was 99 KDa, with polydispersity index values of 2.3 that correspond to those obtained from a conventional radical polymerization reaction. The copolymer presented a unique glass transition temperature (Tg = 54 °C) indicating that no-phase segregation was observed, although a pseudo-block copolymer structure was obtained according to reactivity ratios.

### 3.3. Preparation and Characterization of Self-Assembled Nanoparticles

The aforementioned pseudo-block microstructure and the hydrophobic–hydrophilic balance provided the copolymers with the necessary properties for self-assembling by nanoprecipitation. NPs presented a hydrophobic core mainly formed by covalently linked KT and a hydrophilic shell mainly formed by VI. Nanoprecipitation method was performed as previously described, and NPs were labeled as KT NPs. SEM micrograph of the NPs confirmed the successful NPs’ formation presenting spherical shape, slight polydispersity in size, and diameter of about 100 nm (Appendix A). Regarding hydrodynamic properties, a positive *ξ* value of +30 ± 1 mV was obtained, confirming the presence of VI protonable amine groups on the surface of the NPs and, according to literature, an indication of good stability in suspension [30]. They presented D_h_ of 117 ± 1 nm with low PdI values (0.139 ± 0.004). The spherical morphology, positive surface charge and diameters between 100 and 200 nm made KT NPs suitable for accumulation at inflamed areas [31,32] as well as for an improved sequestration by inflammatory cells avoiding lymphatic drainage [33,34]. The diameter or surface charge of KT NPs did not significantly vary with the different final concentration on the aqueous phase or final volumes under study. Within the studied ranges, the key variable was the concentration of copolymer in the organic phase, which, when increased, led to NPs with 50 nm larger diameter (Table 1). Finally, in order to explore the most suitable conditions for NPs storage, a pH study, stability in suspension study, and freeze-drying study were performed. Figure 2a shows the size distribution and the ξ values of the KT NPs at different pH values (i.e., 4.0, 4.5, 5.0, and 5.5). At pH 5.5, a broadening of the size distribution curve and an increase in the intensity of the peak in the microscale, which might correspond to agglomerated NPs, were observed. A decrease in surface charge was observed as pH increases due to deprotonation of amine groups of VI as the pH approaches to the pK_b_ of VI (pK_b_ = 5.0–6.0). The reduction in the electrostatic repulsion between particles caused NPs aggregation. Therefore, the synthesis was carried out at pH 4 (0.1 M acetic acid) to ensure the protonation of VI and the good stability of the NPs over time. Figure 2b shows that, under these conditions, there were no significant changes in the hydrodynamic properties of KT NPs at 0 days, 14 days, and 28 days when stored at 4 °C. Furthermore, NPs recovered their initial hydrodynamic properties after freeze-drying and dispersion in the buffer solution at pH 4.0 when sonicated for 10 min with an ultrasound tip (30% amplitude) (Figure 2c). Therefore, NPs can be stored in powder or in suspension at pH 4.0 and 4 °C for at least one month.

### 3.4. Dexamethasone and Coumarin-6 Encapsulation

NPs were labeled as XY-KT NPs, with X being the encapsulation efficacy, and Y the drug or dye encapsulated. To achieve the maximum final concentration of Dx in our systems, we performed a screening of different *w*/*w* percentages with respect to the copolymer (5% *w*/*w,* 10% *w*/*w*, 15% *w*/*w,* and 20% *w*/*w*), and the final mass of Dx encapsulated was computed by HPLC (Appendix A). The formulation with 15% *w*/*w* of Dx with respect to the copolymer was chosen for further experiments as it reached the highest Dx %EE and %LC (14% and 3.85%, respectively). The amount of drug encapsulated correlated with an increase in D_h_ of the NPs (140 ± 1 nm) and a reduction of the PdI value (0.081 ± 0.010). However, no significant differences were observed in ξ values (+29 ± 1 mV). Interestingly, for a given concentration, the newly synthesized KT NPs encapsulated twice the amount of Dx than their NAP-bearing homologs [11], which might contribute to an improved anti-inflammatory effect. Coumarin-6 (c6) was used as a fluorescent probe in NPs internalization cellular studies. The dye was encapsulated at a low % *w*/*w* (i.e., 1% *w*/*w*) to avoid fluorescence quenching. Table 2 summarizes the hydrodynamic properties and zeta potential of Dx-loaded, c6-loaded, and unloaded KT NPs.

### 3.5. Uptake Rate of c6-Loaded NPs by RAW264.7 Macrophages

A fast uptake of NPs by inflammatory cells is crucial for retention at inflamed areas [11], and the route by which they are internalized determines their fate inside the cell [35]. Fluorescent c6-loaded NPs were prepared as described before. They were used to monitor NPs internalization by RAW264.7 macrophages over 24 h of exposure at 37 °C (Figure 3a). Figure 3a shows the mass of c6 internalized per cell at different time points (i.e., 1, 2, 4, 6, 8, and 24 h). The uptake rate of NPs was linearly increasing up to 24 h without reaching a plateau, an indication of rapid internalization. However, it was important to differentiate the possible surface adsorption of cationic nanoparticles on the negatively charged cell membrane from actual internalization. To do that, the uptake study was conducted at 4 °C, relying on the decreased membrane recycling occurring at this temperature [36]. The cellular uptake was negligible when incubated at 4 °C in comparison to uptake at 37 °C (Figure 3b), demonstrating the energy-dependent internalization of NPs. The endocytic pathways involved in the cellular uptake of the system were investigated, employing endocytic inhibitors of the main routes used by cationic nanomedicines to enter cells: chlorpromazine (CHL, clathrin-dependent endocytosis), amiloride (AMI, macropinocytosis), and nystatin (NYST, caveolae-mediated endocytosis) (Figure 3b) [35]. When treated with CHL, cellular uptake of KT NPs was reduced by 36 ± 2% (*p* < 0.05) with respect to the untreated positive control (37 °C), respectively. Additionally, pre-treatment AMI led to the reduction of internalization of NPs by 48 ± 3% (*p* < 0.05) relative to the positive control. However, no significant differences were observed after NYST pre-treatment. These results indicated that KT NPs internalization mainly occurred by clathrin-mediated endocytosis (CME) and macropinocytosis. These results correlate with reports of positively charged nanoparticles of ~100 nm diameter predominantly internalized through CME mechanism [37] and those claiming that the electrostatic interactions with the negatively charged cell membrane facilitate cationic molecules internalization through macropinocytosis [23]. Internalization via CME and/or macropinocytosis is recognized to be a more destructive pathway compared to caveolae-mediated endocytosis [23,35]. Vesicle acidification and fusion with lysosomes occurring during CME and macropinocytosis might facilitate the release of both drugs due to the enhanced susceptibility of the ester bond to be degraded at acidic pH and in the presence of esterases. Therefore, the covalent ester bond that links KT to the polymeric backbone will be degraded, and the NPs self-assembled structure will be disrupted.

### 3.6. In Vitro Cytotoxicity Assay of NPs

Cytotoxicity of free Dx (the concentration corresponds to the maximum concentration encapsulated in the NPs), 14Dx-KT NPs, and unloaded KT NPs was tested using murine macrophages (RAW264.7) as model inflammation-related cells. Figure 4 shows the cell viability compared to a control of untreated cells (i.e., non-inflammatory conditions, NIC). None of the concentrations tested were cytotoxic after 24 h, as cell viability was always higher than 70%, and for concentrations above 0.25 mg/mL, it was never below 85% (ISO 10993-5:2009). There were no statistically significant differences (* *p* < 0.05) between Dx-loaded and unloaded NPs (i.e., 14Dx-KT NPs and KT NPs) or between free Dx and 14Dx-KT NPs at any of the tested concentrations.

### 3.7. Effect of Polymeric Nanoparticles Based on Ketoprofen and Dexamethasone on Macrophage NO Levels

Anti-inflammatory capacity of the systems after 24 h and 48 h was assessed by measuring the levels of nitric oxide (NO) released by lipopolysaccharide (LPS)-activated macrophages. When RAW264.7 cells are activated by LPS, they polarize to their pro-inflammatory phenotype (M_1_) and they start overproducing NO, a well-known inflammatory mediator [38]. NO released was measured after 24 h (Figure 5a,b, black) and 48 h (Figure 5a,b, dashed white) of incubation with LPS and either KT NPs (Figure 5a) or 14Dx-KT NPs (Figure 5b) at different concentrations. KT NPs significantly (# *p* < 10^−5^) improved reduced NO release after 48 h when compared to 24 h, whereas NO levels were maintained overtime with 14Dx-KT NPs. This result indicated the faster anti-inflammatory effect of Dx-loaded NPs, which was physically entrapped, allowing an earlier release when compared to the covalently linked NSAID. Moreover, Figure 5c presents LPS-induced NO release after 48 h of LPS administration and challenging with different concentrations of unloaded KT NPs (black) 14Dx-KT NPs (dashed white) or free Dx (red). After 48 h of incubation, all systems counteracted LPS-induced NO release in a statistically significant manner for all concentrations tested (# *p* < 10^–5^) with respect to LPS treated cells (IC,U).

Results demonstrated that NO reduction capacity was maintained after 48 h at any of the concentrations tested in case of KT-based systems. Moreover, Dx-loaded systems performed better than free Dx at specific concentrations of KT-based systems (* *p* < 0.05; [NPs] = 0.045 mg/mL and 0.023 mg/mL). One of these concentrations, 0.045 mg/mL, was chosen for further RT-qPCR analysis as it was the maximum concentration of NPs with improved reduction of NO released levels and it will allow comparison with RT-qPCR results obtained for previously described NAP-based systems. Again, KT-based systems showed an improved behavior when compared to previously described NAP-based systems in terms of reduction of NO released levels at any of the two time points tested, especially Dx-loaded systems [11]. This might be attributed to the higher content in Dx (5.1 µM and 2.55 µM, respectively) and to the faster internalization of KT-based NPs.

### 3.8. Real-Time PCR Analysis of the Expression of M1-M2 Specific Reference Genes after NPs Treatment in Non-Stimulated Macrophages and LPS-Stimulated Macrophages

A high positive surface charge of NPs has been reported as one of the external stimulus leading to M_1_ polarization of macrophages [39,40]. Because of this, the effect of cationic polymeric NPs based on ketoprofen and dexamethasone on M_1_-related gene transcript levels was studied in non-stimulated RAW264.7 cells (NIC) to analyze whether these NPs induce to some extent an M_1_ polarization. Figure 6 shows the transcript levels of *Tnfa*, *Il12b,* and *Il23a* determined by RT-qPCR after 1 day and 7 days of treatment with KT NPs (0.045 mg/mL) and 14Dx-KT NPs (5.1 μM Dx and 0.045 mg/mL NPs), and free Dx (5.1 mM). Results were expressed relative to the corresponding level of expression of each transcript in the untreated sample (i.e., non-inflammatory conditions, NIC). Appendix A shows heat maps and Appendix A numerical RT-qPCR data. Figure 6 shows that there was no significant overexpression of M_1_ markers with respect to the control (NIC) after 1 day or 7 days of treatment with 14Dx-KT NPs; whereas after KT NPs treatment, only *Il12b* was slightly overexpressed after 1 day of treatment. These data demonstrated that, although there was a slight activation of M1 marker genes, there were no significant long-term M_1_-polarization after treatment with cationic KT-based NPs.

Then, in a second set of experiments, the anti-inflammatory effect of the systems was analyzed under inflammatory conditions. RAW264.7 macrophages were cultured in the presence of LPS to mimic pro-inflammatory conditions and treated with either culture media (inflammatory conditions, untreated (IC,U)), free Dx (5.1 μM), unloaded KT NPs (0.045 mg/mL) or 14Dx-KT NPs (5.1 μM Dx and 0.045 mg/mL NPs). The expression of M_1_ and M_2_ (Figure 7) marker genes was presented relative to their corresponding levels in the untreated sample (NIC). Appendix A shows heat maps and Appendix A numerical RT-qPCR data. Regarding M_1_ markers, when compared to basal cellular levels (NIC), LPS-treatment (IC,U) significantly increased transcript levels of *Tnfa* after 1 day of treatment and *Il12b* after 7 days of treatment, and it had no effect on *Il23a.* The short-term induction of *Tnfa* was reversed by all systems tested. However, after 7 days of treatment free of Dx and KT NPs did not affect the expression of *Tnfa,* while the system combining both (14Dx-KT NPs) induced a significant increase in expression of this gene. This might be occurring because of the PGE2 reduction after NSAID/GC combined treatment as both the GC and the NSAID contributed to reducing PGE2 levels [41]. Regarding *Il12b* expression, although 14Dx-KT NPs treatment caused the increase of *Il12b* transcript levels after 1 day of treatment, all systems reversed the LPS-induced overexpression after 7 days of treatment, which was more significant in free Dx and Dx-loaded NPs. Moreover, 14Dx-KT NPs and KT NPs induced a slightly significant overexpression of *Il23a* with respect to the LPS-treated control that was reversed after 7 days of treatment. Therefore, in the long term, *Tnfa* overexpression was accompanied by normal levels of *Il12b* and *Il23a.* These findings correlate with the described dual role of *Tnfa*: as an initiator of inflammatory response early in the infection and selective regulator of inflammatory response in the later stages of inflammation [42].

Regarding M_2_ markers, LPS-treatment induced *Vegfa* and *Il10* expression in the short term, while it inhibited it after 7 days of treatment. Moreover, *Tgfb1* expression was not affected in the short term but significantly repressed after 7 days of treatment. This long-term effect on *Tgfb1* expression was significantly reversed by all treatments. On the contrary, the addition of 14Dx-KT NPs produced a significant increase in *Il10*, *Tgfb1,* and *Vegfa* expression with respect to +LPS conditions at both time-points tested. Interestingly, all systems maintained the overexpression of *Il10* in time, something that did not occur in simulated inflammatory conditions. *Il10* is considered the most important anti-inflammatory cytokine in humans [43], confirming the anti-inflammatory capacity of the systems. These data indicated that KT NPs presented a potent anti-inflammatory behavior improving that of the previously described NAP-based analog. This result was expected as KT is a more potent NSAID than NAP in terms of COX inhibition [21,22]. Finally, regarding the synergistic *Il12b,* gene repression was not observed, meaning that the combination of Dx with a stronger COX-inhibitor did not imply a more potent synergistic effect. This result suggested that the COX inhibition was not contributing to the synergistic *Il12b* gene repression in murine macrophages when combining NAP and Dx. Moreover, this synergistic repression was observed for the NAP/Dx system even in non-LPS-activated macrophages in which COX was not overexpressed. All these data indicate that the COX-dependent route does not play a key role in the synergistic effect observed, and further studies on the mechanism of action should focus on MAPK pathways and 5-LO metabolism as there are reports of modulation of *Il12b* expression in antigen-presenting cells (i.e., macrophages and dendritic cells, mainly) through these pathways [13,14,15,16,17].

## 4. Conclusions

Anti-inflammatory NPs that combine KT (covalently attached) and Dx (physically entrapped) were successfully prepared, and their biological activity was evaluated. The amount of Dx entrapped by KT-bearing polymer drugs into NPs was optimized by testing different %Dx (*w*/*w*, with respect to the copolymer) in such a way that the amount of Dx encapsulated was maximized. Moreover, it was demonstrated that the spherical shape, the hydrodynamic properties, and positive surface charge of KT NPs (D_h_ = 100–200 nm, PdI below 0.2, and ξ close to +30 mV) translated into rapid internalization by macrophages through CME and macropinocytosis, two internalization routes that might facilitate the enzymatic and pH-mediated release of KT and Dx in the lysosome. 14Dx-KT NPs and unloaded KT NPs reduced LPS-induced NO release when compared to the control, with the Dx-loaded system more effective at 24 h due to the early release of the physically entrapped Dx. Moreover, the unloaded KT NPs were able to reduce to almost basal levels (NIC) the expression of all M_1_ markers tested. The RT-qPCR analysis also concluded that the synergistic inhibition of *Il12b* was not occurring in a significant manner for 14Dx-KT NPs. However, long-term treatment with 14Dx-KT NPs led to *Tnfa* overexpression and regulation of *Il12b* and *Il23a* expression, reducing it up to normal cellular levels. As a conclusion, the KT-based systems described in this paper present interesting anti-inflammatory activity as they reduced NO released levels and M1 markers expression while increasing M2 markers expression under inflammatory conditions. Moreover, KT-based systems were rapidly internalized by macrophages, which might favor the retention at inflamed areas through the ELVIS effect. Therefore, they could be used by themselves, as well as encapsulating Dx or other hydrophobic drugs, for the treatment of inflammatory processes. Moreover, it was demonstrated that the use of a KT, an NSAID with more potent COX-inhibitory activity did not cause synergistic repression of *Il12b* gene when combined with Dx. This, together with the fact that this synergistic repression was observed for the NAP/Dx system even in non-LPS-activated macrophages in which COX was not overexpressed [11], indicated that the COX-dependent route was not crucial for the synergy observed. Therefore, further studies should be done on NAP/Dx mechanism of action focusing on MAPK pathways and 5-LO metabolism.

## Figures and Tables

**Figure 1 pharmaceutics-12-00723-f001:**
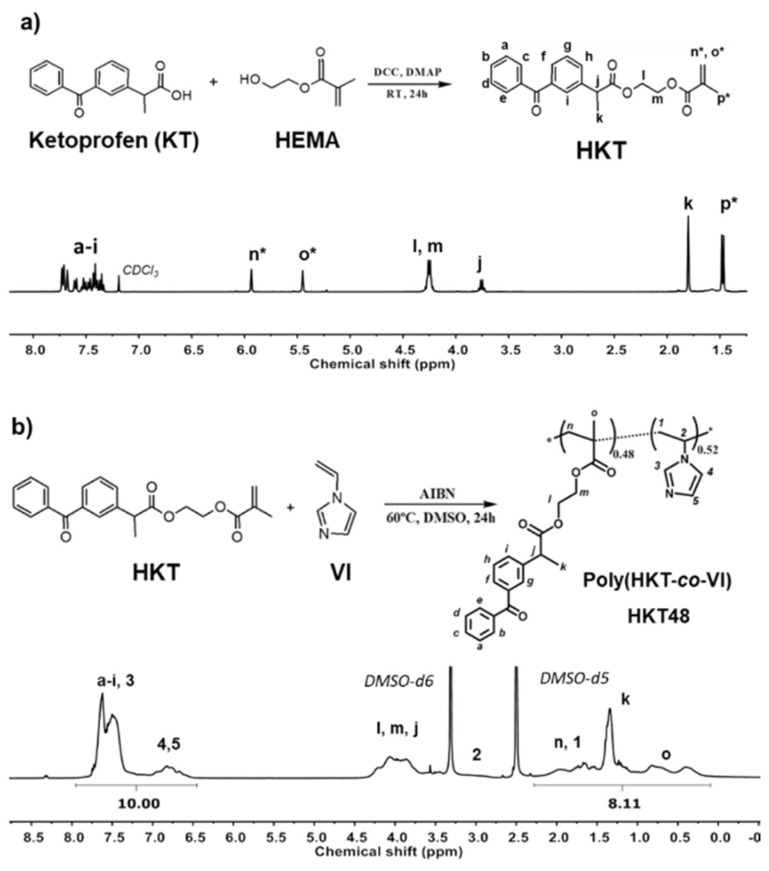
(**a**) Synthesis and ^1^H-NMR spectra of the methacrylic derivative of ketoprofen (HKT), and (**b**) synthesis via free radical copolymerization and ^1^H-NMR spectra of poly(HKT-co-VI) copolymer. Note: n * and o * refer to the cis and trans protons of the carbonyl group.

**Figure 2 pharmaceutics-12-00723-f002:**
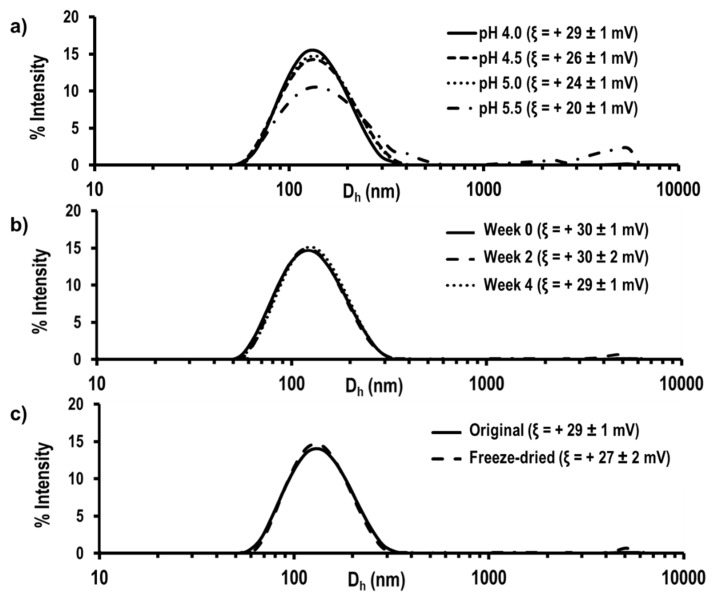
Size distribution (by intensity) and z-potential (ξ) of ketoprofen (KT) NPs measured (**a**) at different pH values or at pH 4.0; (**b**) at day 0 and after 28 days stored at 4 °C; and (**c**) before freeze-drying (dashed line) and after freeze-drying and tip sonication for 10 min.

**Figure 3 pharmaceutics-12-00723-f003:**
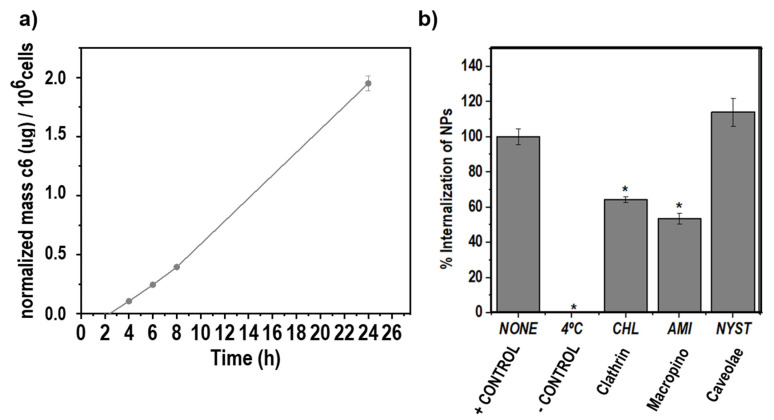
(**a**) Normalized mass of coumarin-6 (c6) internalized per cell at different time points for 47c6-KT NP, and (**b**) %internalization of 47c6-KT NPs at 37 °C (NONE, positive control), 4 °C (negative control), and at 37 °C after pre-treatment with chlorpromazine (CHL), amiloride (AMI), and nystatin (NYST) (*: significant difference compared to NONE, * *p* < 0.05).

**Figure 4 pharmaceutics-12-00723-f004:**
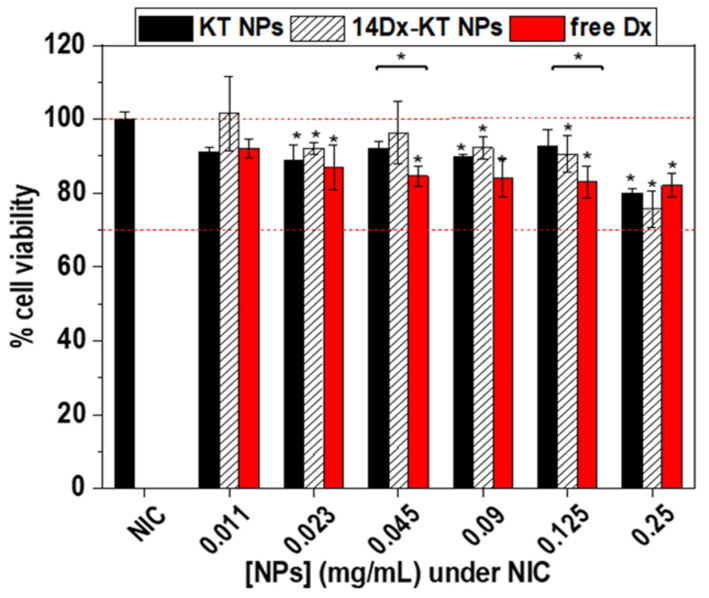
Cell viability of RAW264.7 macrophages treated with different concentrations of unloaded KT NPs (black), 14Dx-KT NPs (dashed white), or free Dx (red) over 24 h. The diagrams include the mean, the standard deviation (n = 8), and the ANOVA results (* *p* < 0.05 statistically significant difference with cells under non-inflammatory conditions (NIC) and between the different systems).

**Figure 5 pharmaceutics-12-00723-f005:**
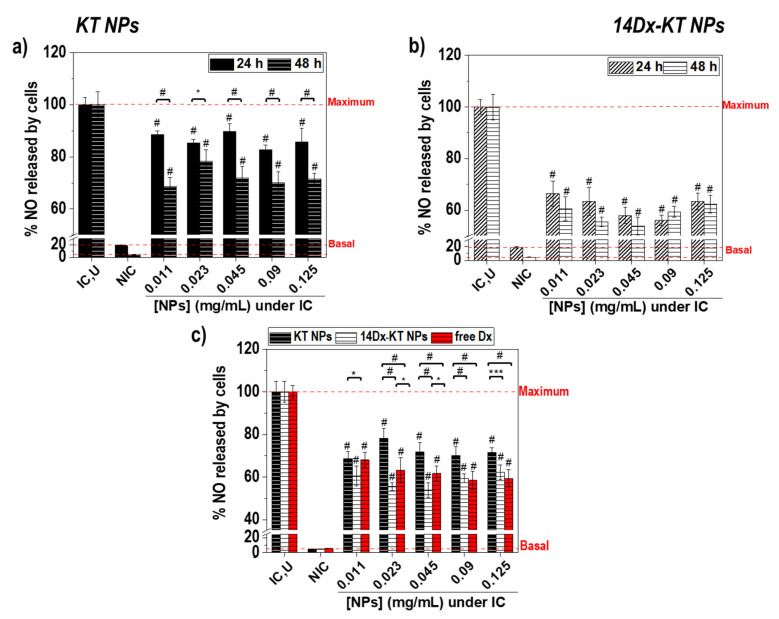
NO release by RAW264.7 macrophages after 24 h (black) and 48 h (dashed white) with no treatment (NIC), treatment with LPS alone (IC,U), and treatment with LPS and different concentrations of (**a**) unloaded KT NPs and (**b**) 14Dx-KT NPs; (**c**) NO release after 48 h with no treatment (NIC), treatment with LPS (IC,U), and treatment with LPS and different concentrations of loaded NAP NPs or KT NPs (black), 8Dx-NAP NPs or 14Dx-KT NPs (dashed white), or free Dx (red). The diagrams include the mean, the standard deviation (n = 8), and the ANOVA results (* *p* < 0.05, *** *p* < 10^−3^, and # *p* < 10^−5^ statistically significant difference with IC,U cells or between 24 h and 48 h time points).

**Figure 6 pharmaceutics-12-00723-f006:**
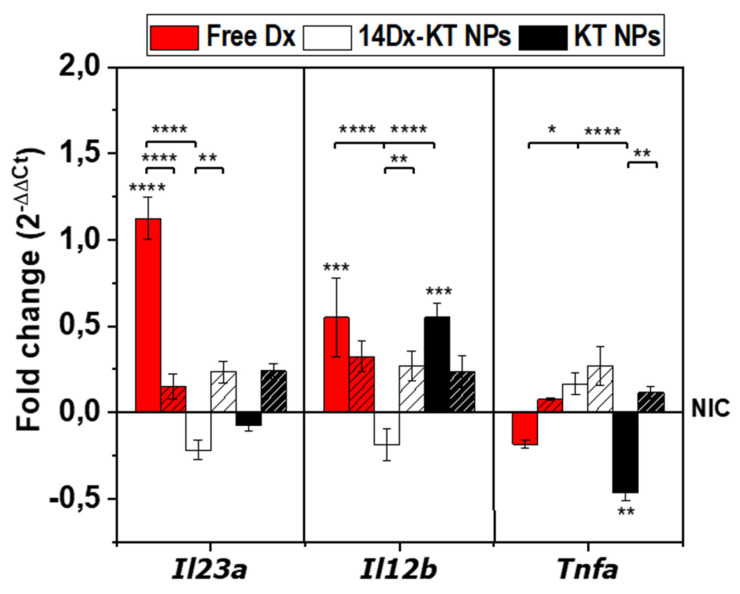
Quantitative real-time PCR data. Graphical presentation of gene transcript levels of M_1_ markers in non-LPS-activated samples (non-inflammatory condition, NIC) treated with free dexamethasone (free Dx, 5.1 μM, red), Dx-loaded ketoprofen-bearing NPs (14Dx-KT NPs, 5.1 μM Dx, and 0.045 mg/mL NPs, white) and unloaded ketoprofen-bearing NPs (KT NPs, 0.045 mg/mL, black) for 1 day (plain) and 7 days (dashed). Results are expressed relative to the corresponding level of expression of each transcript in the untreated sample. The diagrams include the mean, the standard deviation (n = 2), and the ANOVA results (*—comparison with untreated NIC control, * *p* < 0.05, ** *p* < 0.01, *** *p* < 0.001 and **** *p* < 0.0001).

**Figure 7 pharmaceutics-12-00723-f007:**
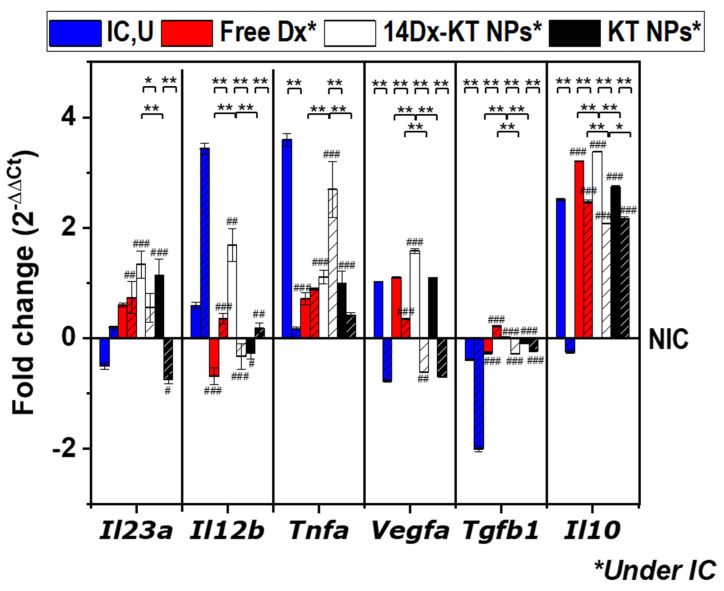
Quantitative real-time PCR data. Graphical presentation of gene transcript levels of M_1_ markers (*Il23a, Il12b, and Tnfa*) and of M_2_ markers (*Vegfa, Tgfb1, and Il10*) in samples treated with LPS (500 ng/mL) and either treated with culture media (IC,U, blue), free dexamethasone (free Dx, 5.1 μM, red), Dx-loaded ketoprofen-bearing NPs (14Dx-KT NPs, 5.1 μM Dx and 0.045 mg/mL NPs, white), and unloaded ketoprofen-bearing NPs (KT NPs, 0.045 mg/mL, black) for 1 day (plain) or 7 days (dashed). Results were expressed relative to the corresponding level of expression of each transcript in non-inflammatory conditions (RAW264.7 alone, NIC). The diagrams include the mean, the standard deviation (n = 2), and the ANOVA results (#—comparison with IC,U control, # *p* < 0.05, ## *p* < 0.01, and ### *p* < 0.001; *—comparison between systems, * *p* < 0.05 and ** *p* < 0.01).

**Table 1 pharmaceutics-12-00723-t001:** Hydrodynamic properties (i.e., hydrodynamic diameter (D_h_), polydispersity of size distribution (PdI), and zeta potential (ξ)) of KT NPs at a different final concentration of nanoparticles in the aqueous phase ([NPs]_A.P._) and at different final volumes (V_F_).

[NPs]_A.P._ ^a^	V_F_ (mL)	[HKT48]_O.P._ ^b^	D_h_ ^c^ (nm)	PdI ^d^	ξ ^e^ (mV)
1.0 mg/mL	10	10 mg/mL	117 ± 1	0.139 ± 0.004	+29 ± 1
	20	10 mg/mL	119 ± 3	0.153 ± 0.018	+29 ± 1
	30	10 mg/mL	116 ± 1	0.142 ± 0.027	+30 ± 1
5.0 mg/mL	10	10 mg/mL	115 ± 1	0.124 ± 0.008	+30 ± 1
	20	10 mg/mL	119 ± 1	0.112 ± 0.013	+31 ± 1
	30	15 mg/mL	163 ± 3	0.186 ± 0.021	+29 ± 1

^a^ Final concentration of nanoparticles (NPs) in the aqueous phase, ^b^ concentration of copolymer in the organic phase, ^c^ mean hydrodynamic diameter. and ^d^ polydispersity of the size distribution obtained by dynamic light scattering (DLS), ^e^ mean zeta potential obtained by laser Doppler electrophoresis (LDE).

**Table 2 pharmaceutics-12-00723-t002:** Summary of encapsulation efficiency (%EE), loading capacity (%LC), and hydrodynamic properties (i.e., mean hydrodynamic diameter (D_h_), polydispersity of size distribution (PdI), and zeta potential (ξ)) of newly prepared KT NPs, 14Dx-KT NPs, and 47c6-KT NPs.

Sample	% EE ^a^	% LC ^b^	D_h_ ^c^ (nm)	PdI ^d^	ξ ^e^ (mV)
**KT NPs**	-	-	117 ± 1	0.139 ± 0.004	+30 ± 1
**14Dx-KT NPs**	14	3.85	140 ± 1	0.081 ± 0.010	+29 ± 1
**47c6-KT NPs**	47	0.47	126 ± 2	0.154 ± 0.021	+32 ± 2

^a^ Percentage of encapsulation efficiency of coumarin-6, **^b^** percentage of loading capacity of coumarin-6, ^c^ mean hydrodynamic diameter, and ^d^ polydispersity of the size distribution obtained by DLS, ^e^ mean zeta potential obtained by LDE.

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
