# Peer review of "Anti-Inflammatory Polymeric Nanoparticles Based on Ketoprofen and Dexamethasone"

_pharmaceutics, 2020, doi:10.3390/pharmaceutics12080723_

Round 1
Reviewer 1 Report
This manuscript tested the combination of ketoprofen and dexamethasone combination in a nanoparticle in which ketoprofen along with methacrylic acid formed the nanoparticle whereas dexamethasone was incorporated in these nanoparticles. The experimental details and methods are clearly described. As such the combination is interesting, there are some gaps in this work. Please provide answers for following questions and suggestive improvements.
- Please check the grammatic mistakes, although they are les they are still present.
- The release of both drugs from nanoparticles is not probed. Although cell culture models provide some insight into release, they are complicated and several factors interfere with interpretation. An in vitro release test is needed since authors are making lot of time-dependent biological conclusions in this work.
- Please move c6-loaded nanoparticle preparation to main paper.
- The major issue with this paper is figures
- organic mixture. Change 8:2 to “80:20”?
- Lines 180-183: “Supernatant was discarded and cell’s pellet was lysate with ethanol to release and dissolve internalized c6. Fluorescence of supernatant was measured (458/540nm, excitation/emission)..” Was the supernatant discarded or measured?
- What is 14Dx?
- Line 265 - Please check figure s1 is not the microstructure.
- Line 290-293: “Furthermore, NPs recovered their initial hydrodynamic properties after freeze-drying and dispersion in the buffer solution at pH 4.0 when sonicated for 10 minutes with an ultrasound tip (30% amplitude) (figure 2c). Therefore, NPs can be stored in powder or in suspension at pH 4.0 and 4 ºC for at least one month.”
Sonication for re-dispersion is not a viable option for commercial scale and use in the hospital setting! Also, I do not agree with the storage recommendation, please justify.
- Why more extreme pH not tested for size? Particles will be in the blood at pH: 7.4, what will be the behavior?
- Figure 5 and 6. Need better explanation. Move heat maps to supplemental. Present clear graphs with clear explanations. It seems that authors tried to explain all the observed changes with some phenomenon. Combine the figures if possible and report results. The M1 and M2 study needs clear explanation.
12. In general KT-Dx NPs did not show more improvement compared to free Dx probably due to release issues. In vivo studies will shed more light.
Author Response
Reviews for pharmaceutics-847599
Title: Anti-inflammatory polymeric nanoparticles based on ketoprofen and dexamethasone
Author(s): Eva Espinosa-Cano, Maria Rosa Aguilar*, Yadileiny Portilla, Domingo F. Barber, Julio San Román
Reviewers’ Comments to Author:
Reviewer: 1
This manuscript tested the combination of ketoprofen and dexamethasone combination in a nanoparticle in which ketoprofen along with methacrylic acid formed the nanoparticle whereas dexamethasone was incorporated in these nanoparticles. The experimental details and methods are clearly described. As such the combination is interesting, there are some gaps in this work. Please provide answers for following questions and suggestive improvements.
- Please check the grammatic mistakes, although they are less they are still present.
Reply: We appreciate the comment and we have carefully checked and corrected grammar mistakes along the manuscript.
- The release of both drugs from nanoparticles is not probed. Although cell culture models provide some insight into release, they are complicated and several factors interfere with interpretation. An in vitro release test is needed since authors are making lot of time-dependent biological conclusions in this work.
Reply: We completely agree with the reviewer, the release kinetics of the drugs are not presented in the manuscript. We have tried to monitor the in vitro release of ketoprofen and dexamethasone at pH 7.4 and 37°C. However, we faced several difficulties during the experiments: first, the release was neglected in PBS at pH 7.4 and 37°C during 7 week. We added esterases to the PBS in order to force the enzymatic release of ketoprofen and disassembling of the NPs. Both drugs were released but the release profile completely depended on esterase concentration and enzyme refreshing. Therefore, the obtained release profile was not representative of the real release conditions and decided not to include them in the manuscript. We consider that the release of both drugs will take place after cell internalization. NPs will enter the lysosome/endosome were the high concentration of esterases together with the acidic pH will allow a faster and efficient release of the drug. According to this, instead of a ketoprofen release study we considered more appropriate to perform a study of the uptake of the NPs by macrophages, which indirectly is a study of the ketoprofen release.
- Please move c6-loaded nanoparticle preparation to main paper.
Reply: According to the reviewer’s comment, we have moved the c6-loaded nanoparticles preparation and characterization to main paper.
- The major issue with this paper is figures
Reply: According to the reviewer’s comment, we have improved all figures.
- organic mixture. Change 8:2 to “80:20”?
Reply: According to the reviewer comment, changes were done in line 153, 159 and 161
“8:2” was changed to “80:20”
- Lines 180-183: “Supernatant was discarded and cell’s pellet was lysate with ethanol to release and dissolve internalized c6. Fluorescence of supernatant was measured (458/540nm, excitation/emission)..” Was the supernatant discarded or measured?
Reply: We thank the reviewer for his observation, to clarify this issue we have made changes in lines 180-183.
“After further centrifugation at 10,000 rpm, fluorescence of supernatant was measured (458/540 nm, excitation/emission) by a Multi-Detection Microplate Reader Synergy HT (BioTek Instruments; Vermont, USA).”
- What is 14Dx?
Reply: According to the suggestion of the reviewer, we have included the explanation in line 165 before we use 14Dx terminology.
“NPs were labeled as XY-KT NPs being X the encapsulation efficacy, Y the drug or dye encapsulated.”
- Line 265 - Please check figure s1 is not the microstructure.
Reply: We thank the reviewer for his comment and we agree with him. We have decided to eliminate figure S1 in line 265.
- Line 290-293: “Furthermore, NPs recovered their initial hydrodynamic properties after freeze-drying and dispersion in the buffer solution at pH 4.0 when sonicated for 10 minutes with an ultrasound tip (30% amplitude) (figure 2c). Therefore, NPs can be stored in powder or in suspension at pH 4.0 and 4 ºC for at least one month.”
Sonication for re-dispersion is not a viable option for commercial scale and use in the hospital setting! Also, I do not agree with the storage recommendation, please justify.
Reply: We agree with the reviewer, although it was an interesting result, as we have never recovered initial hydrodynamic properties after freeze-drying, it is not viable at a commercial scale. However, we described the best storage conditions of NPs in laboratory conditions as we have demonstrated that they were stable up to 1 month in suspension and we have demonstrated that it is possible to recover their hydrodynamic properties by sonication.
- Why more extreme pH not tested for size? Particles will be in the blood at pH: 7.4, what will be the behavior?
Reply: We thank the reviewer for his comment. With the pH study we just wanted to establish the storage pH when the NPs were in aqueous suspension and, because of that, we did not considered necessary to test higher pH values. However, as the reviewer mentioned, the physiological pH is about 7.4 and, hence, we needed to solve this problem for in vitro testing. As previously reported (DOI: 10.1039/C4CS00487F (Review Article) Chem. Soc. Rev., 2015, 44, 6287-6305), the proteins in FBS are able to stabilize cationic NPs (electrostatic stabilization) for in vitro testing and these proteins will be present in blood too. According to that, we diluted our nanoparticles at different volume ratios NPs:culture media choosing 1:5 as the one that stabilize the NPs.
- Figure 6 and 7. Need better explanation. Move heat maps to supplemental. Present clear graphs with clear explanations. It seems that authors tried to explain all the observed changes with some phenomenon. Combine the figures if possible and report results. The M1 and M2 study needs clear explanation.
Reply: We thank the reviewer for the observation and, according to it, we have improved the PCR discussion and we have combined figure 7a and figure 7b to facilitate discussion.
- In general KT-Dx NPs did not show more improvement compared to free Dx probably due to release issues. In vivo studies will shed more light.
Reply: We do agree with the reviewer and we are planning to do further in vivo studies in the future

Reviewer 2 Report
Comments:
- In line 156 and Supplementary, the authors mention filtering Dx-KT NPs through 1 μm Nylon filters to remove remaining insoluble Dx, but did not mention the manufacturer for these filters. Also, shouldn’t this size allow the less-than-200nm-in-diameter NPs slip through?
- A clarification about the timeline of the PCR experiment could elucidate whether the cells were lysed immediately after the 24 hours of direct contact with the NPs or at 24 hours after removing the NPs. Also, was the cells’ media periodically refreshed in the case of the 7 days experiment (lines 212-230)?
- In line 493, while there was a reduction of specific cytokines overexpression, it was investigated only in monocultured macrophages and not in tissues (in vitro or in vivo models). So maybe it would be more appropriate to conclude that the cytokines were reduced to normal cellular levels.
- Since the biological study was performed using only in vitro monocultured macrophages - so there was no experiment using more complex in vivo inflammation models or even in vitro ones - the conclusion concerning the “the KT-based systems […] were rapidly internalized by macrophages favoring the retention at inflamed areas through the ELVIS effect” (lines 493-497) seems an overstatement.
- While there are data available about the characteristics concerning the hydrodynamic properties (Dh), polydispersity of size distribution (PdI) and zeta potential (ξ) for KT NPs and even 47c6-KT NPs, those concerning 14Dx-KT NPs are missing (Table S1).
- Is there an explanation for the decrease of total dexamethasone encapsulated in KT-NPs, as identified by HPLC, for 20% Dx with respect to poly(HKT-co-VI) copolymer, while the final quantity of dexamethasone retrieved from 5, 10 and 15% Dx with respect to poly(HKT-co-VI) copolymer showed a constant, linear increase? (Figure S3)
Suggestions:
- The underlined degree symbol is recognized as "Masculine Ordinal Indicator" by MS Office, so it should be changed to the classic "degree" symbol.
- I recommend rephrasing the title of result 3.8. (lines 402-403).
- There are some inconsistencies regarding dexamethasone encapsulation between Materials and methods (mentioning 5, 10, 15, 20 and 25%; line 151) to Results (line 308) and Supplementary information (mentioning only 5, 10, 15 and 20%).
- It is not clear early on whether the c6 containing NPs are KT NPs or not. (lines 175-183).
- The first reference about KT NPs carrying dexamethasone is made in line 206, by mentioning “14Dx-KT NPs” but not explaining the acronym (“14” before the “Dx-KT NPs”) – the possible explanation could be identified later, in line 311.
- Concerning Table 1, the last row shows the data corresponding to the final NPs concentration of 5 mg/ml, but the copolymer concentration in organic phase is different (15 mg/ml) from the rest of the shown values (10 mg/ml). Is it possible to include the hydrodynamic properties for the 10 mg/ml copolymer concentration in organic phase, so that the table has some symmetry between the two final NPs concentrations in aqueous phase?
- Figure 4 caption does not include “CP” found in the chart and what the acronym means. Also, the title of the OX axis <[NPs] (mg/mL)> has the inferior margin cut off.
- Figure 5: could different colors/shading for the different NPs (KT NPs and 14Dx-KT NPs) be used constantly across the three charts so that the results are more accessible to follow? Also, the “#” is not explained in the caption.
- Figure 6: there is no “#” in chart 6a, but it is mentioned in the caption.
- For Figures 5 and 7, the "*" and "#" symbols are very blurry.
- Final proofreading should resolve various grammar errors and phrasing inconsistencies (e.g:
-
Line 19: drugs at inflamed tissue -> drugs in the inflamed tissue
-
Line 22: pathways to favor its retention -> pathways to favor their retention
-
Line 47: nanoparticles [...] has -> nanoparticles [...] have
-
Line 180: pellet was lysate -> pellet was lysed
-
Line 271: a positive ξ values of + 30 ± 1 -> a positive ξ value of + 30 ± 1 mV
-
Line 288: to assure the protonation -> to ensure the protonation
-
Line 481: its biological activity -> their biological activity
-
Line 445: consequence of the PEG2 reduction... is it PEG2 or PGE2?)
-
Author Response
Reviews for pharmaceutics-847599
Title: Anti-inflammatory polymeric nanoparticles based on ketoprofen and dexamethasone
Author(s): Eva Espinosa-Cano, Maria Rosa Aguilar*, Yadileiny Portilla, Domingo F. Barber, Julio San Román
Reviewers’ Comments to Author:
Reviewer: 2
Comments:
- In line 156 and Supplementary, the authors mention filtering Dx-KT NPs through 1 μm Nylon filters to remove remaining insoluble Dx, but did not mention the manufacturer for these filters. Also, shouldn’t this size allow the less-than-200nm-in-diameter NPs slip through?
Reply: We have added the manufacturer information in the text (line 156 and supplementary). Moreover, in this step we chose a pore size of 1um because it will eliminate unsoluble crystalized Dx that was not encapsulated without changing the size distribution of the NPs.
- A clarification about the timeline of the PCR experiment could elucidate whether the cells were lysed immediately after the 24 hours of direct contact with the NPs or at 24 hours after removing the NPs. Also, was the cells’ media periodically refreshed in the case of the 7 days experiment (lines 212-230)?
Reply: We agree with the reviewer and we have change the paragraph to clarify this point. Regarding the CM refreshment, we have changed the media every two days.
“Culture medium was refreshed every 48 h. Total RNA was extracted from cells after 1 and 7 days of NPs addition. PureLink RNA Mini Kit (Applied Biosystems) was used for this purpose following manufacturer's instructions.”
- In line 493, while there was a reduction of specific cytokines overexpression, it was investigated only in monocultured macrophages and not in tissues (in vitro or in vivo models). So maybe it would be more appropriate to conclude that the cytokines were reduced to normal cellular levels.
Reply: according to the reviewer’s comment, we have changed the word “tissue” by “cellular”
- Since the biological study was performed using only in vitro monocultured macrophages - so there was no experiment using more complex in vivo inflammation models or even in vitro ones - the conclusion concerning the “the KT-based systems […] were rapidly internalized by macrophages favoring the retention at inflamed areas through the ELVIS effect” (lines 493-497) seems an overstatement.
Reply: we agree with the reviewer and we have rephrased the sentence.
“As a conclusion, the KT-based systems described in this paper present interesting anti-inflammatory activity as they reduced NO released levels and M1 markers expression while increasing M2 markers expression under inflammatory conditions. Moreover, KT-based systems were rapidly internalized by macrophages, which might favor the retention at inflamed areas through the ELVIS effect.”
- While there are data available about the characteristics concerning the hydrodynamic properties (Dh), polydispersity of size distribution (PdI) and zeta potential (ξ) for KT NPs and even 47c6-KT NPs, those concerning 14Dx-KT NPs are missing (Table S1).
Reply: In order to clarify this points, we have included hydrodynamic properties and zeta potnetial of c6-loaded nanoparticles in the main text together with 14Dx-KT NPs and KT NPs data (table 2)
- Is there an explanation for the decrease of total dexamethasone encapsulated in KT-NPs, as identified by HPLC, for 20% Dx with respect to poly(HKT-co-VI) copolymer, while the final quantity of dexamethasone retrieved from 5, 10 and 15% Dx with respect to poly(HKT-co-VI) copolymer showed a constant, linear increase? (Figure S3)
Reply: This behavior has already been described by other authors (Adsorption Science & Technology 32(5) 2014: DOI: 10.1260/0263-6174.32.5.365; PLoS ONE 9(12) 2014: e113558. DOI:10. 1371/journal.pone.0113558). Accoridng to the literature, this might be a matter of Dx solubility in water. At higher concentrations, Dx might by less soluble and when it reaches the aqueous solution it may crystalize limiting its encapsulation.
Suggestions:
- The underlined degree symbol is recognized as "Masculine Ordinal Indicator" by MS Office, so it should be changed to the classic "degree" symbol.
Reply: We have changed all underlined degree symbol by the classic one.
- I recommend rephrasing the title of result 3.8. (lines 402-403).
Reply: As recommended we have changed the title of this section
“Real-time PCR analysis of the expression of M1-M2 specific reference genes after NPs treatment in non-stimulated macrophages and LPS-stimulated macrophages”
- There are some inconsistencies regarding dexamethasone encapsulation between Materials and methods (mentioning 5, 10, 15, 20 and 25%; line 151) to Results (line 308) and Supplementary information (mentioning only 5, 10, 15 and 20%).
Reply: we thank the reviewer for the comment and we have changed 5, 10, 15, 20 and 25% to 5, 10, 15 and 20% in materials and methods
- It is not clear early on whether the c6 containing NPs are KT NPs or not. (lines 175-183).
Reply: to clarify that we have changed the title of this section.
“Uptake rate of c6-loaded KT NPs by RAW264.7 macrophages”
- The first reference about KT NPs carrying dexamethasone is made in line 206, by mentioning “14Dx-KT NPs” but not explaining the acronym (“14” before the “Dx-KT NPs”) – the possible explanation could be identified later, in line 311.
Reply: According to the suggestion of the reviewer, we have included line 311 explanation in line 165.
“According to this, NPs were labeled as XY-KT NPs being X the encapsulation efficacy, Y the drug or dye encapsulated.”
- Concerning Table 1, the last row shows the data corresponding to the final NPs concentration of 5 mg/ml, but the copolymer concentration in organic phase is different (15 mg/ml) from the rest of the shown values (10 mg/ml). Is it possible to include the hydrodynamic properties for the 10 mg/ml copolymer concentration in organic phase, so that the table has some symmetry between the two final NPs concentrations in aqueous phase?
Reply: We do agree with the reviewer. However, to prepare 5 mg/mL NPs at a concentration of 10 mg/mL in the organic phase we needed to have an organic volume larger than the aqueous one. In this conditions, the NPs did not formed.
- Figure 4 caption does not include “CP” found in the chart and what the acronym means. Also, the title of the OX axis <[NPs] (mg/mL)> has the inferior margin cut off.
Reply: As suggested, we have changed CP in the figure by CM and we have improved the OX axis appereance.
- Figure 5: could different colors/shading for the different NPs (KT NPs and 14Dx-KT NPs) be used constantly across the three charts so that the results are more accessible to follow? Also, the “#” is not explained in the caption.
Reply: We have changed the colors to make the figures easier to follow and we have included # in the caption.
“#p < 10-5 statistically significant difference with LPS+ cells or between 24 h and 48 h time points).”
- Figure 6: there is no “#” in chart 6a, but it is mentioned in the caption.
Reply: We agree with the reviewer and we have eliminated # from the caption
- For Figures 5 and 7, the "*" and "#" symbols are very blurry.
Reply: We have improved the resolution of the figure.
- Final proofreading should resolve various grammar errors and phrasing inconsistencies (e.g:
- Line 19:drugs at inflamed tissue -> drugs in the inflamed tissue DONE
- Line 22:pathways to favor its retention -> pathways to favor their retention DONE
- Line 47:nanoparticles [...] has -> nanoparticles [...] have DONE
- Line 180:pellet was lysate -> pellet was lysed DONE
- Line 271:a positive ξ values of + 30 ± 1 -> a positive ξ value of + 30 ± 1 mV DONE
- Line 288:to assure the protonation -> to ensure the protonation DONE
- Line 481:its biological activity -> their biological activity DONE
- Line 445:consequence of the PEG2 reduction... is it PEG2 or PGE2?) DONE

Round 2
Reviewer 1 Report
The authors made some requested changes however major points are not addressed. Specifically comment # 2 and # 10.
Comment 2: I am not satisfied with the authors’ response. It is imperative that release be evaluated in vitro. Hoping for an in vitro or in vivo release is not sufficient and makes this system less reliable for future optimizations, and in general to critically understand its performance. Therefore, authors are requested to include this information in supplemental at least.
Comment 10: Results for pH 7.4 stability should be reported.
Author Response
According to the editor's instructions, we made changes to address comment #10.
Comment 10: Results for pH 7.4 stability should be reported.
Reply: According to the reviewer’s comment, we have included data of NPs stability in DMEM (figure S6 and figure S7). Hydrodymanic properties were measured 0 h and 24 h after dilution because 24 h was the longest time that we incubated cells with the NPs. In all in vitro experiments, after 24h, we substituted NPs suspension with fresh media. The figures show that NPs are stabilized by FBS at dilutions with media 1:3, 1:4 and 1:5 (NPs:DMEM; v/v) being 1:5 the one at which the size and the PdI presented smaller values.
